# No Longer Underappreciated: The Emerging Concept of Astrocyte Heterogeneity in Neuroscience

**DOI:** 10.3390/brainsci10030168

**Published:** 2020-03-13

**Authors:** Francisco Pestana, Gabriela Edwards-Faret, T. Grant Belgard, Araks Martirosyan, Matthew G. Holt

**Affiliations:** 1Laboratory of Glia Biology, VIB-KU Leuven Center for Brain and Disease Research, 3000 Leuven, Belgium; francisco.pestana@kuleuven.vib.be; 2Laboratory of Neuronal Wiring, VIB-KU Leuven Center for Brain and Disease Research, 3000 Leuven, Belgium; gabriela.edwards@kuleuven.vib.be; 3Life & Medical Sciences (LIMES) Institute, Faculty of Mathematics and Natural Sciences, University of Bonn, 53115 Bonn, Germany; 4The Bioinformatics CRO, Niceville, FL 32578, USA; grant@grantbelgard.com; 5Leuven Brain Institute, KU Leuven, 3000 Leuven, Belgium

**Keywords:** astrocyte, heterogeneity, morphology, development, molecular profiling, physiology, function, injury and disease, injury and disease

## Abstract

Astrocytes are ubiquitous in the central nervous system (CNS). These cells possess thousands of individual processes, which extend out into the neuropil, interacting with neurons, other glia and blood vessels. Paralleling the wide diversity of their interactions, astrocytes have been reported to play key roles in supporting CNS structure, metabolism, blood-brain-barrier formation and control of vascular blood flow, axon guidance, synapse formation and modulation of synaptic transmission. Traditionally, astrocytes have been studied as a homogenous group of cells. However, recent studies have uncovered a surprising degree of heterogeneity in their development and function, in both the healthy and diseased brain. A better understanding of astrocyte heterogeneity is urgently needed to understand normal brain function, as well as the role of astrocytes in response to injury and disease.

## 1. Introduction

Astrocytes comprise the largest class of glial cells in the mammalian brain, and are essential for central nervous system (CNS) development and function [1]. Compared to their neuronal counterparts, however, astrocytes remain understudied, largely due to the lack of tools allowing specific astrocyte labeling and manipulation. Hence, while neurons are known to display extensive molecular and functional diversity even within brain regions [2,3], astrocytes have traditionally been discussed as a homogeneous cell type [1], despite the wide range of key CNS functions that astrocytes are thought to participate in. These include processes as diverse as promoting synapse formation, maintaining synaptic homeostasis (through the control of extracellular K^+^ levels and removal of neurotransmitters), modulation of synaptic transmission, formation and maintenance of the blood-brain-barrier, control of cerebral blood flow and immune response [1]. This diversity of functions raises the obvious question of whether they are performed by all astrocytes, as presupposed, or whether functional subsets of astrocytes exist.

In a key review in 2010 [4], Ye Zhang and Ben Barres argued that astrocyte heterogeneity was an underappreciated topic in neurobiology. These authors made a compelling argument that a better understanding of astrocyte heterogeneity is necessary if we are to fully understand CNS function in both the healthy and injured (or diseased) brain. In the intervening period, substantial methodological improvements have led to an increased focus on the issue of astrocyte heterogeneity. As such, evidence is now accumulating which supports the importance of astrocyte heterogeneity throughout the CNS, both between and within brain regions. We hope that this review acts as a ‘primer’, quickly and clearly illustrating some of the emerging concepts in astrocyte heterogeneity, using selected examples in the areas of anatomy, developmental biology, molecular profiling, cell physiology, function and response to injury and disease. The majority of work described will have been conducted in rodents (mice and rats), unless otherwise stated, because of their wide use in astrocyte research. Where possible, we attempt to build on Zhang and Barres, with a focus on recent studies, particularly those that have used tissue slices, or better still in vivo models, to avoid the caveats associated with reduced complexity systems [5]. Readers are directed to excellent recent reviews to obtain a more complete view of the field [6,7,8].

## 2. Morphological Heterogeneity

Astrocytes were first identified as a heterogeneous cell type based on gross morphology and differential expression of common marker proteins, such as the intermediate filament GFAP and glutamate transporter GLAST [7,9]. The most common division, based on morphology, is between protoplasmic astrocytes found in the grey matter and fibrous astrocytes found in the white matter (Figure 1). Additional morphological types in the mouse CNS include tanycytes, radial cells, Bergmann glia, velate glia, marginal glia, perivascular glia and ependymal glia [10]. Müller glia are a specialist astrocyte type found in the retina [1]. Multiple astrocyte morphologies may be found in the same brain region [9]. Generally speaking, individual astrocytes are thought to occupy distinct, non-overlapping spatial locations in the tissue (so-called tiling), with only minimal overlap between the ends of astrocyte processes [11], although a recent report using clonal analysis suggests that up to 20% of astrocytes show considerable spatial overlap in cortex [12]. Astrocyte density can vary considerably between regions [10,13], although individual cell counts always depend on the histological methodologies applied, which needs to be considered when comparing studies.

Protoplasmic astrocytes have an instantly recognizable morphology. They exhibit a spongiform profile, consisting of a central soma and four to ten major processes, which in turn branch off into thousands of smaller processes [8]. The tips of astrocyte processes ensheath synapses and blood vessels [1]. At the synapse, there is compelling evidence for bidirectional communication between neurons and astrocytes, with astrocytes thought to modulate synaptic transmission—a concept known as the ‘tripartite synapse’ [14]. Recent careful examination of cells using high-resolution fluorescence imaging has indicated subtle differences between protoplasmic astrocytes occupying distinct brain regions [15,16]. A systematic study of mouse hippocampal and striatal astrocytes, using iontophoresis of Lucifer Yellow dye, revealed that astrocytes in these regions have equivalent somatic volumes, number of main processes and total cell volumes. However, striatal astrocytes generally occupy a larger territory than hippocampal astrocytes and associate with more neuronal cell bodies, while hippocampal astrocytes associate with a larger number of excitatory synapses (an observation which was further confirmed with serial block face scanning electron microscopy) [15]. Such morphological heterogeneity also extends to astrocytes occupying the same brain region. The morphology of cortical protoplasmic astrocytes was investigated using cells outlined using a membrane tagged TdTomato combined with cytoskeletal GFP-labeling. Four prominent morphological subtypes were identified (based on consideration of multiple parameters including cell elongation, flatness and orientation), which distribute through the various cortical layers in differing proportions [16].

However, it is wrong to think that astrocyte morphology is fixed. Historically, there has been ample evidence that astrocyte processes are capable of changing their morphology (summarized by Theodosis and colleagues [17]). However, due to technical limitations associated with classical immunostaining and electron microscopy, such studies have tended to focus on either specialist synapses, such as the suprachiasmatic nucleus during circadian rhythmicity [18], or cortical synapses after prolonged periods of activity [19]. Recent advances in real-time imaging, using the astrocyte-specific dye sulforhodamine 101 (SR101) [20] or astrocyte-specific GFP expression [21], in both tissue slices and in vivo now suggest that modification of astrocytes processes can occur rapidly, in response to neuronal activity in mouse hippocampus and somatosensory cortex. Structural remodeling of this sort will have important consequences for astrocytic regulation of synaptic transmission, for example through the modulation of glutamate transporter proximity to synapses. However, this view of dynamic astrocyte processes has been recently challenged by in vivo experiments using overexpression of genetically encoded FRET sensors, which suggest that interactions between neurons and astrocytes are stable in the striatum [22]. The reasons for this discrepancy are unclear, but may be related to intrinsic differences in the biology of the hippocampus, cortex and striatum, differences in the methodologies used, or a combination of both.

There is now accumulating evidence that astrocyte processes exist as compartmentalized signaling units. In mature astrocytes, overexpression of a FLAG-tagged variant of the glutamate transporter GLT1 in combination with an anti-FLAG antibody conjugated to a quantum dot, allowed the movement of the transporter to be tracked [23]. Experiments show that GLT1 diffuses across the surface of the astrocyte process, becoming ‘trapped’ at synaptic sites, in order to remove glutamate from the synaptic cleft and terminate synaptic transmission. Despite the use of harsh solubilization conditions, GLT1 was isolated from brain extracts in a complex containing glycolytic enzymes and mitochondria [24], indicating that this complex is extremely stable when formed. In theory, such a complex provides a mechanism to spatially match energy production to transporter activity. It is tempting to link these two observations and speculate that capture of GLT1 into such a complex underlies its retarded movement. However, the proof for such a hypothesis remains lacking. Indeed, what could be the trigger for GLT1 capture? It is known that astrocytes respond to the activity of single synapses with compartmentalized Ca^2+^ signaling [25,26], and that mitochondrial positioning at synapses is dependent on neuronal activity and Ca^2+^ signaling [27,28]. Hence, the effects of Ca^2+^ in processes may play a central role in GLT1 capture.

Whether process remodeling is dependent on de novo protein synthesis or the redistribution/reuse of existing proteins is unclear at present. However, astrocyte processes contain mRNAs [29] and glial remodeling is blocked by inhibitors of protein synthesis, such as anisomycin [30]. Therefore, it appears likely that local translation (perhaps in an activity-dependent manner [31]) will emerge as playing a critical role.

Hence, although we now have a good understanding of basic astrocyte morphology throughout the brain, it remains unclear how key signaling machinery is distributed through the cell and how this distribution changes, if at all, upon astrocyte activity. Although technically demanding, experiments aimed at elucidating the composition of astrocyte processes are likely to be particularly revealing, given that these structures are in close proximity to synapses and blood vessels. Future technical improvements, such as (live) super-resolution imaging [32], combined with the ability to monitor mRNA trafficking [33] and local protein synthesis [34] will undoubtedly aid such investigations.

## 3. Developmental Heterogeneity

Since astrocyte heterogeneity was first recognized, a major question in the field has been how it is established. Is it set at the level of individual progenitor populations, or does local environment play a role? In reality, strong evidence exists for both possibilities.

Embryonic astrocytes derive from radial glia [35,36,37] and several lines of evidence indicate that astrocyte specification in the spinal cord and forebrain originates from a segmental template, induced by the action of known morphogens, such as sonic hedgehog (SHH), bone morphogenetic proteins (BMPs) and Wnt proteins, which induce the expression of specific transcription factors in discrete cell populations (comprehensively summarized by Ben Haim and Rowitch [7]). For example, the basic helix-loop-helix proteins Olig2 and SCL regulate oligodendrocyte versus astrocyte production in the p2 domain of the spinal cord [38], while the transcription factor Emx1 regulates astrocyte production in the cortical plate and corpus callosum [39].

Radial glia are no longer present in the post-natal mouse brain. Despite this, there is a rapid expansion in the number of cortical astrocytes during early post-natal development. Experiments combining retroviral labeling of dividing cells with GFP and immunostaining for astrocyte-specific BLBP (brain lipid binding protein) demonstrated that in cortical layers I -IV this expansion is due to the local proliferation of ‘pioneer’ astrocytes generated from radial glia [36], producing ‘clouds’ of clonally related cells [40]. However, the final distribution of astrocytes in the adult largely matches the original trajectory of their radial glia precursors [37,39], implying that astrocytes do not migrate tangentially and may retain spatial information at later developmental times, which is presumably crucial for the formation of the mature CNS. In fact, neither stab wound injury [39,41] nor toxin-mediated cell ablation [39] appear sufficient to overcome this strict regional allocation and induce mass migration of astrocytes to the injury site.

However, it appears that not all astrocyte functions are hardwired during development. In the cerebellum at least, neurons control astrocyte diversity through the release of SHH [42]. Crucially, Bergmann glia and velate astrocytes show considerable transcriptomic divergence, demonstrating the power of local signaling cues in determining fate. Whether this is a cerebellum specific mechanism remains to be determined. Certainly, the structure of the cerebellum with clearly demarcated cell locations and cell morphologies facilitates such studies. In contrast, cortical astrocytes are grossly similar at the anatomical level and large numbers of cells in the tissue share a common developmental lineage (see above). However, SHH is also produced in adult mouse cortex and SHH receptors are expressed on a proportion of cortical astrocytes [43]. This implies that the influence of local neuronal activity may be more subtle in cortex, with recent single cell data suggesting that local synaptic activity may influence astrocyte identity at the transcriptomic level (see Section 4). Of course, this does not preclude a contribution for other signaling pathways, such as fibroblast growth factor (FGF) signaling [15,44]. A potential role for pathways commonly associated with development is interesting, as evidence suggests that persistent activation of these pathways is necessary to maintain astrocyte diversity [45]. In this respect, a role for glutamatergic neuronal signaling in astrocyte development is not surprising, with genetic ablation of neuronal VGLUT1 adversely affecting post-natal development of cortical astrocytes [46]. It is also required for expression of the potassium channel Kir4.1 in astrocytes of the ventral horn of the spinal cord [47]. Furthermore, genetic manipulations affecting cortical layering, for example neuronal deletion of *Dab1* [16] or *Satb2* [48], also influence astrocyte positioning.

Hence, it appears cues from neighboring cells in the CNS cooperate with early patterning events to promote astrocyte diversity from a single progenitor.

## 4. Molecular Heterogeneity

In our opinion, the largest recent insights into astrocyte heterogeneity have been driven by experiments using transcriptome wide sequencing to reveal the underlying molecular machinery of the cell.

In a seminal study of cell type heterogeneity, Doyle and colleagues [49] introduced the TRAP (translating ribosome affinity purification technique) and showed considerable differences between the transcriptomes of cortical astrocytes, cerebellar astrocytes and cerebellar Bergmann glia.

However, it is wrong to think of the transcriptome as static. Using astrocyte-TRAP, Boisvert and colleagues [50] analyzed the transcriptome in cortex, hypothalamus and cerebellum and found region specific, aging-related changes (consistent with previous reports [51,52,53]). The fact that expression of mRNAs associated with synapse function decreased and expression of mRNAs associated with immune response increased suggests that astrocytes become pro-inflammatory during aging (see also a complementary study by Clarke et al. [54]), perhaps predisposing certain individuals to develop neurodegenerative disease [55] (as discussed in Section 7).

Morel and colleagues [56] performed a similar study, including hippocampus, cortex, nucleus accumbens, caudate putamen, thalamus and hypothalamus. They found a high degree of overlap between genes expressed in the cortex and hippocampus, while the nucleus accumbens, caudate putamen and hypothalamus showed expression of unique mRNAs not expressed in astrocytes from any other brain region studied. Amongst the most interesting transcripts identified were those for Hevin (*Sparcl1*) and SPARC (*Sparc*). Hevin is a pro-synaptogenic protein [57] and its mRNA was expressed at similar levels across all regions. SPARC, on the other hand, antagonizes Hevin action [57] and its mRNA was enriched in thalamic and hypothalamic astrocytes. Results were interpreted as a gradient of astrocyte gene expression along the dorsal-ventral axis of the brain, with consequences for excitatory synapse formation. Interestingly, the Boisvert study [50] also found evidence for changes in gene expression across the motor, somatosensory and visual cortices, indicative of a rostral-caudal gradient.

A further example of a gene expression gradient can be found in the work comparing hippocampal and striatal astrocytes [15]. These show differences in transcriptome (using TRAP) and proteome (using semi-quantitative liquid chromatography-tandem mass-spectrometry: LC-MS/MS) [15]. Interestingly, validation of the proteomic results with immunohistochemistry demonstrated that the protein µ-crystallin is differentially expressed amongst striatal astrocytes, occupying a gradient along the dorsal-ventral axis [15]. The functional consequences of this expression gradient are unknown at present.

Although the TRAP and MS experiments described here report data from whole astrocyte populations, these studies consistently indicate a complex picture of astrocyte gene expression with multiple axes of variation. Interestingly, even subtle changes in gene expression can have functional consequences: using cultured neurons and astrocytes in a ‘mix-and-match fashion’ revealed that astroglial modulation of synaptic maturation is region-selective [56].

The use of tissue microdissection and intersectional labeling strategies has contributed greatly to our understanding of intra-regional heterogeneity, particularly in cortex. Isolation of astrocytes from upper (2/3 and 4) and deeper (5 and 6) layers of cortex revealed differences in gene expression between the two sets of astrocytes [16,48]. These included genes involved in synapse formation and elimination, such as *Sparc* and *Merkt* [16] and *Chrdl1* and *Il33* [48] amongst others. A partial GLT1/EAAT2 promoter fragment drives TdTomato expression in a subgroup of astrocytes in cortical layer 5 [58], which express the synaptogenic protein Norrin (and see Section 7). A double reporter line (GLT1/EAAT2-TdTomato: Aldh1l1-eGFP) identifies three distinct classes of cortical astrocytes [59], with unique transcriptomes, tissue locations and electrophysiological characteristics. Staining with antibodies specific for CD51, CD63 and C71 in the Aldh1l1-eGFP mouse identifies five astrocyte subpopulations present in cortex, olfactory bulb, brain stem, thalamus, cerebellum and spinal cord [60], albeit in differing proportions. Interestingly, these populations vary in their synaptogenic potential.

In terms of defining the true extent of heterogeneity in a cell population, single cell methods provide, at least in theory, the truest measure. To date, such studies have generally reported a low degree of heterogeneity within specific brain regions. Gokce and colleagues [61] were unable to identify clear, distinct subtypes in the striatum, reporting a continuous transcriptional gradient defined by a unimodal population distribution, albeit with clear separation of genes encoding neurotransmitter transporters and glutamate receptors from those encoding ribosomal proteins and cell polarity regulators. This is in contrast to the proteomics-based findings from Chai and colleagues [15], which defined two distinct populations based on µ-crystallin expression. Zeisel and colleagues [62] identified 2 astrocyte subtypes in the somatosensory cortex and hippocampus, distinguished by differential expression of *Gfap* (which maps as expected to the pial layer) and *Mfge8* (which extends through the remainder of the cortex). A more extensive follow up study by the same group [3] largely recapitulated these findings, with 7 astrocyte subtypes identified across the entire CNS. In another study, Saunders and colleagues [2] identified 8 distinct astrocyte types across 9 major brain regions.

At face value, these studies suggest that the number of discrete astrocyte subtypes in the CNS is rather low, certainly in comparison to reported neuronal diversity. However, the Gokce study [61] does raise the pertinent issue of how cell types are defined computationally, as opposed to identification of cell states. Furthermore, these studies have typically used a ‘one size fits all’ protocol, in which neither the tissue dissociation methods employed or the protocols used for sequencing library preparation have been optimized for a specific cell type, raising questions about whether astrocytes are subject to negative selection, based on issues of low viability after isolation, low mRNA content, etc. To investigate the issue of astrocyte diversity in more depth, our lab recently developed a protocol specifically for use with adult astrocytes and paired it with deep sequencing [63]. This work identified 5 distinct subtypes in cortex and hippocampus; two largely distinct groups identified as putative progenitors and three closely related subtypes likely to represent mature astrocytes. These subtypes contained markers identified in previous single cell studies, albeit over a wider number of astrocyte types, suggesting that the optimized protocol used in our study allows for a more subtle cell characterization. Interestingly, even though two of the ‘mature’ subtypes were largely intermixed in the cortex, they showed differential expression of transcripts involved in synapse formation and modulation of synaptic transmission, suggesting a fine degree of control over local CNS function. In our opinion, more targeted sequencing of astrocytes will also reveal a greater degree of heterogeneity across other brain regions.

Our findings of high levels of heterogeneity arising from subtle differences in transcript expression are further supported by a study from Bayraktar and colleagues [48], who took our single cell data set and by mapping back gene expression profiles to known layer specific markers, using large-scale fluorescence in situ hybridization (ISH), found evidence for both ‘laminar’ and ‘non-laminar’ gene expression in the dataset. Interestingly, laminar gene expression was offset from that classically defined by neuronal wiring patterns [48] and varied along both the dorsal-ventral and rostral-caudal axes.

Hence, the majority of evidence supports the fact that astrocytes are heterogeneous at the transcriptome level. There is evidence both for discrete patterns of gene expression restricted to specific brain regions, as well as for subtle gradients of gene expression spanning regions. Astrocytes may well share common genes, such as those regulating essential metabolic processes, but it appears there is sufficient variation in transcript expression to produce unique astrocyte subtypes, which are specialized to perform certain specific functions. As it stands, however, the majority of current single cell studies show considerable differences in the mouse lines used, age of mouse used, brain regions sampled, isolation method, number of astrocytes isolated, library preparation method employed and sequencing depth (summarized in [63]). These technical factors make it challenging to link subtypes identified by individual studies. This is an obstacle to overcome by systematic studies with the goal of building a comprehensive picture of astrocyte diversity across development and brain regions.

Finally, although the amount of data produced in single cell sequencing experiments can seem overwhelmingly complex, studies on single genes, such as *Sema3a* [64] and *Kcnj10* [47] in the spinal cord (discussed in Section 6), have shown the power of gene expression data to inform hypothesis driven science. The development of new tools utilizing such data (for example ISH probes, antibodies, viral vectors and mouse lines) will undoubtedly be needed to aid future investigations.

## 5. Physiological Heterogeneity

Historically, astrocytes have been of little interest from an electrophysiological perspective, as they do not show the rapid fluctuations in membrane potential typical of action potential firing in neurons [65]. Generally speaking, mature astrocytes are characterized by a hyperpolarized resting potential that is set close to the equilibrium potential for K^+^ (E_K_) (approximately −80 mV), low input resistance (5−20 MΩ) and a linear current to voltage relationship, indicating a relative lack of voltage-dependent ionic conductances in the plasma membrane [65].

Fluctuations in membrane potential are tightly linked to changes in neuronal activity. Indeed, due to their strong, inward K^+^ conductance, astrocytes are highly sensitive to changes in extracellular K^+^ levels that are associated with neuronal activity, meaning they play a key role in regulating neuronal firing [66]. Interestingly, a recent comprehensive comparison of hippocampal and striatal astrocytes [15] found that they differ significantly in the size of their Ba^2+^-sensitive K^+^ currents. This suggests regional specialization, consistent with the fact that the striatum comprises predominantly GABAergic neurons with hyperpolarized membrane potentials, unlike the hippocampus which contains primarily glutamatergic neurons [15], meaning that striatal astrocytes have a lower requirement for K^+^ buffering and K^+^ dissipation. As expected, the difference in K^+^ conductance was also reflected in differential expression of genes encoding K^+^ channels, consistent with transcriptome informing function [15].

It has long been recognized that astrocytes can exist in extensive gap junction coupled networks (syncytia), mediated primarily by Connexins 30 and 43 [67]. The extent of gap junction coupling differs widely between brain regions [68,69] and is age dependent [70]. The connectivity of astrocyte networks can also be highly specific, with extensive gap junction coupling in the barrel cortex occurring between astrocytes in the same barrel, but limited coupling occurring between astrocytes from different barrels [71]. Gap junction coupling confers isopotentiality on a syncytium [72], minimizing depolarization due to elevated levels of local extracellular K^+^ and thereby maintaining a sustained driving force for highly efficient K^+^ uptake and maintenance of local homeostasis. Interestingly, although hippocampus and striatum show similar levels of gap junction coupling (approximately 100 coupled cells per network), the gap junction blocker carbenoxolone has much more profound effects in striatum [15], arguing for molecular heterogeneity in gap junctions [73] and suggesting caution when using this drug in future experiments. Independent of its function as an ion channel, Connexin 30 has also been reported to play a crucial role in regulated cell adhesion and migration [74], with genetic ablation experiments demonstrating a role for the protein during insertion of astrocyte processes into synaptic clefts. Crucially, Connexin 30 levels in astrocytes are controlled by neuronal activity, which effectively sets up a feedback loop to regulate synaptic transmission [75].

Astrocytes respond to various stimuli with increases in intracellular Ca^2+^ [6]. Such signaling has been intensively studied, as it is generally thought to represent the predominant form of cellular communication [76]. Generally, studies have been descriptive, with few reports of absolute Ca^2+^ levels in resting and activated astrocytes. Such measurements would undoubtedly be informative and contribute to our understanding of astrocyte function. Furthermore, technical limitations associated with the use of small organic indicators meant that historically measurements were usually limited to the cell soma [77]. Even at this level, differences in Ca^2+^ signals were reported, with marginal astrocytes in cortical layer 1 showing frequent asynchronous Ca^2+^ transients and astrocytes in layers 2/3 showing infrequent Ca^2+^ transients [78]. Recent studies, however, have taken advantage of the fact that genetically encoded calcium indicators (GECIs) can be expressed in a cell type specific manner and targeted to specific subcellular locations. They have revealed a rich diversity of Ca^2+^ signals in cells, ranging from the generation of signaling microdomains [79,80,81] through to global waves that encompass entire astrocytes, including their cell bodies [81,82]. Ca^2+^ responses may be spontaneous [79,83] or evoked by neuronal activity [25,26], with differences between brain regions reported; striatal and hippocampal astrocytes show differences in these two forms of Ca^2+^ signal [15]. Reported Ca^2+^ sources include channel-mediated Ca^2+^ entry [84], release from IP_3_R_2_-dependent intracellular stores [81,85] and release from mitochondria [86].

Crucially, a focus on intracellular Ca^2+^ elevations as activity signals may have artificially constrained our view of astrocyte function, as dopamine appears to both elevate and lower astrocyte Ca^2+^ in the stratum radiatum, depending on the signaling mechanisms used, Ca^2+^ source and the activity of extrusion mechanisms [87]. It is also important to note that oscillations in intracellular Ca^2+^ appear superimposed on a heterogeneous resting Ca^2+^ landscape within the cell [88]. Rat hippocampal astrocytes in acute tissue slices show a gradient of resting Ca^2+^ from the cell soma to their processes. This gradient is age dependent and astrocytes with high and low basal Ca^2+^ occupy contiguous space [88]. The level of resting Ca^2+^ is influenced by many factors, including Ca^2+^ leak from stores or through channels, the presence of fixed and mobile Ca^2+^ buffers, and extrusion mechanisms [89]. Hence, heterogeneity in the basal Ca^2+^ reflects the Ca^2+^ homeostasis machinery in cellular subcompartments, which will ‘sculpt’ the dynamics of evoked Ca^2+^ oscillations. Hence, Ca^2+^ signaling should no longer be considered a uniform blunt signal, but in all likelihood subtly encodes information in its dynamics. The introduction of next generation signal processing tools, capable of accurately describing the dynamics of (subcellular) Ca^2+^ transients, will undoubtedly help in revealing the subtleties of this signaling pathway [90].

However, a significant question remains unanswered: what is the function of these differing types of Ca^2+^ signal in astrocytes? It is tempting to speculate that they play functionally distinct roles. Interestingly, striatal and hippocampal astrocytes show differences in direct GPCR-mediated signaling (activated using DREADDs; Designer Receptors Exclusively Activated by Designer Drugs), with the Gi/o pathway more efficient in promoting Ca^2+^ transients in striatal astrocytes, suggesting differential expression of intracellular signaling cascades—although differences in DREADD expression levels and/or coupling into downstream signaling pathways cannot be completely excluded, due to the use of viral vectors for DREADD delivery [15]. This is further reinforced by the finding that expression of the intermediate early gene, *c-fos*, is greater in striatal astrocytes than hippocampal astrocytes following the activation of Gi coupled DREADD [15]. Cortical astrocytes show differential responses to the metabotropic agonist phenylephrine, which reflect the distribution of morphologically and transcriptomically distinct astrocyte populations [63]. Finally, work from our own lab suggests that local synaptic activity as well as the behavioral state of the animal are crucial in shaping intracellular Ca^2+^ signaling in the mouse visual cortex, suggesting that astrocytes act as signal integrators to produce distinct physiological effects [82]. This is in agreement with previous work showing the effects of the neuromodulator noradrenaline on astrocyte signaling [91].

Ca^2+^ mediated activities may include controlling cerebral blood flow [92] and modulating synaptic transmission [25], possibly through the Ca^2+^-mediated release of signaling molecules. Beyond Ca^2+^, the use of genetically encoded indicators for cAMP [93] and lactate [94] has revealed transient changes in the intracellular concentrations of these molecules under defined conditions, providing another layer of complexity to astrocyte signaling. It is likely that a combination of different signaling modalities, with different spatio-temporal properties, will control the myriad of functions performed by these cells.

## 6. Functional Heterogeneity

Perhaps the single most important advantage of astrocyte diversity is that it enables the creation of specialized neuron-glia units, which can drive complex behaviors. Multiple examples of distinct astrocyte subpopulations critical to key CNS functions have now been uncovered.

A comprehensive comparison of astrocytes isolated from the dorsal and ventral horns of the mouse spinal cord revealed distinct transcriptomic differences between the two populations [64]. In particular, ventral astrocytes were enriched in *Sema3a* transcripts, which encode an axon guidance cue that binds to the neuropilin 1 receptor on motor neurons [64]. Gene ablation experiments demonstrated a crucial role for astrocyte *Sema3a* in orientation and survival of α-motor neurons during development [64]. Similarly, ventral horn astrocytes are also enriched in transcripts for *Kcnj10*, which encode for the K^+^ channel Kir4.1 [47]. Unlike *Sema3a*, expression of *Kcnj10* is dispensable for motor neuron survival [47]. However, loss of *Kcnj10* expression in astrocytes does have non-cell autonomous effects on local neurons: fast α-motor neurons are smaller than wild type counterparts and show dramatically different electrophysiological profiles. These deficits correlate with reduced muscle size and decreased muscle strength [47] (Figure 2). Crucially, *Kcnj10* levels appear reduced in astrocytes generated from amyotrophic lateral sclerosis (ALS) patients [47], providing a potential explanation for the muscle weakness seen in this disease [95] (see also Section 7.1).

Once formed, specialist circuits control a number of critical behaviors in the adult mouse. For example, astrocytes in the suprachiasmatic nucleus (SCN) drive the molecular oscillations underpinning circadian behavior in mammals, controlling clock gene expression in neurons via glutamatergic signaling [96]. Astrocytes also act as rhythm generators in the ventral brain stem area, responding to physiological decreases in pH with the release of ATP, which activates chemoreceptor neurons to induce adaptive changes in breathing [97]. They are also key for generating rhythmic activity in the rat trigeminal sensorimotor circuit responsible for mastication during feeding [98]. Insulin signaling in hypothalamic astrocytes is a key component in glucose metabolism, with the genetic ablation of insulin receptors in mice leading to hyperphagia and impaired regulation of systemic glucose levels [99]. Finally, homeostatic control of dopamine levels by astrocytes in the prefrontal cortex is essential for synaptic transmission and plasticity, playing a crucial role in memory and behavioral flexibility [100]. Likewise, the activation of astrocytes (but not neurons) in hippocampal CA1, using a Gq-coupled DREADD, was shown to enhance memory acquisition in mice [101].

Crucially, however, functional heterogeneity may not be restricted to region-specific astrocytes. Dopamine receptor D1 and dopamine receptor D2 containing medium spiny neuronal subtypes are intermixed in the dorsal striatum [102]. When stimulated, endocannabinoid release from these neurons leads to increases in intracellular Ca^2+^ in different subsets of striatal astrocytes [102]. Crucially, Ca^2+^ uncaging experiments in these specific astrocytes evoked a transient synaptic potential only in homotypic pairs of synaptically connected neurons [102].

Taken together, these examples provide clear proof that specialized astrocyte subsets are responsible for the function of specific neuronal circuits, and are capable of synapse-specific regulation. Using the increasingly comprehensive gene expression data now available, it is likely that further examples of astrocyte specialization will be uncovered in the immediate future. However, given the complex transcriptional ‘fingerprints’ which define astrocyte subtypes [48,63], sophisticated intersectional strategies (e.g., split-Cre systems) will be needed for specific labeling and manipulation [103].

## 7. Heterogeneity in Injury and Disease

Astrocytes are involved in all neuropathologies, ranging from acute injury through to chronic neurodegenerative conditions. However, how astrocytes are involved in these different pathologies remains unclear and needs to be addressed. Even the basic issue of whether astrocytes are primary initiators or modifiers of disease is debatable, and probably depends on the condition in question [6]. As it increasingly appears that all injuries and diseases elicit a unique astrocytic response, a comprehensive description of astrocyte involvement in all conditions is beyond the scope of this work. Rather, in this section, we have chosen specific examples which we think best illustrate key emerging concepts relating to astrocyte heterogeneity in injury and disease. However, for readers interested in specific conditions, in depth reviews can generally be found (for example Alexander’s disease [104], Alzheimer’s disease (AD) [105] and ischemic injury [106]).

### 7.1. Astrocyte-Specific Diseases

At the time of writing, only Alexander’s disease has been categorically demonstrated to be a primary astrocyte disease, due to mutations in the *Gfap* gene, which is specifically expressed in astrocytes [107]. More than 50 mutations have been reported and associated with development of the disease. Although it shows variable age of onset, being defined as ‘infant’ (0–2 years), ‘juvenile’ (2–12 years) or ‘adult’ (>12 years), a common phenotype is the formation of Rosenthal fibers in astrocytes, thought to be indicative of cell stress and loss of internal cell homoeostasis [104]. The destruction of myelin characteristic to the disease [108] typically relates to the age of disease onset and matches the regional distribution of GFAP expressing astrocytes in the CNS [109], which are found in large numbers in the white matter and provide metabolic support to myelinating oligodendrocytes [104].

Recent evidence now suggests that Norrie disease, which is caused by mutations in the *Ndp* (Norrie disease pseudoglioma) gene located on the X-chromosome, may be linked to the secretion of active Norrin protein from distinct astrocyte subsets and subsequent activation of the Wnt signaling pathway in surrounding cells [58]. Consistent with the involvement of Wnt signaling in CNS patterning [110] and synapse formation [111], patients generally present with blindness. Studies in a mouse model carrying a mutation in exon 2 of the *Ndp* gene [112], which results in the deletion of 56 amino acids from the N-terminus of the protein, suggest that this is due to disorganization of the ganglion cell layer and loss of the outer plexiform layer in the retina. Approximately 30%–50% of males with Norrie disease also present with intellectual disability, behavioral abnormalities, or psychotic-like features [113], consistent with the abnormal dendrite patterning and spine formation, impaired synaptic transmission and behavioral abnormalities seen in the Norrin knockout mouse [58,114].

Expression of Kir4.1 is heavily enriched in astrocytes associated with motor neurons in the ventral spinal cord. Reduced levels of Kir4.1 lead to a specific reduction in the size of fast twitch α-motor neurons and loss of muscle strength [47] (Figure 2; see above). Amyotrophic lateral sclerosis (ALS) is a degenerative disease which mainly affects the lower motor neurons in the brainstem and ventral horn of the spinal cord, with fast twitch α-motor neurons most severely affected [115]. ALS is familial in approximately 10% of cases with mutations in the superoxide dismutase 1 (*SOD1*) gene found in 20% of affected families [116]. Interestingly, astrocytes derived from ALS patients with a SOD1^D90A^ mutation showed a cell autonomous reduction in Kir4.1 levels [47]. Although there was no evidence for enhanced loss of motor neurons in a SOD mutant mouse model following genetic ablation of Kir4.1, this study suggests that the initial clinical presentation of weakness in ALS may be related to astrocyte-specific down-regulation of Kir4.1.

### 7.2. Astrocyte Response to External Injury or Disease

Generally speaking, injury or disease is associated with morphological changes and upregulation of classical markers in astrocytes, such as GFAP, a process known as reactive gliosis [117]. In the case of severe insults, such as paralysis-inducing spinal cord injury resulting from spinal cord crush [118] or lesion [119], the consequences are major tissue remodeling and the formation of a prominent glial scar. The glial scar derives almost entirely from newly proliferated astrocytes with elongated shape, whose processes interdigitate forming a barrier (scar) around the injury site [117]. The role of the glial scar is heavily debated and appears double-edged. On the one hand, it is thought the scar acts to seal off the damaged area immediately following injury to prevent the escape of toxic species, such as oxygen free radicals and proteases released from damaged cells, which would otherwise spread through the CNS causing damage [117]. Unfortunately, however, the changes underlying scar formation are effectively irreversible; as regenerating axons do not pass through the glial scar, reinnervation of affected areas and functional recovery is effectively prevented [117]. The basis of this negative effect was long thought to be the expression of inhibitory chondroitin sulfate proteoglycans (CSPGs) by scar forming astrocytes [117]. However, this view has recently been challenged in a spinal crush model [118], where the specific targeting of reactive astrocytes, either by genetic inactivation of the key regulatory protein STAT3 or through cell ablation using diphtheria toxin, failed to stimulate axon regrowth and actually compromised tissue integrity. Surprisingly, in this study, levels of inhibitory CSPGs were maintained in the environment of the scar in the absence of reactive gliosis; in fact, transcriptome sequencing of reactive astrocytes revealed that these cells actually express CSPG types supportive of axonal growth [118]. The reason for the discrepancy between studies is unclear, but may be due to the sustained expression of inhibitory CSPGs from other cells types (e.g., fibroblasts and pericytes) which are minor components of the glial scar or differential response to injury type (as discussed below).

In other injuries, such as stab wound models in the hippocampus [120] and cortex [41], morphological changes are more moderate, with cells generally displaying varying degrees of hypertrophy but maintaining the tiling arrangement seen in healthy tissue [6]. While studies using classical immunohistochemistry-based techniques have revealed important aspects of the injury response, by definition they represent a snap-shot of the tissue at a fixed time point. Advances with in vivo imaging now allow the whole injury response to be evaluated over time. Crucially, the response to cortical stab wound injury appears to be heterogeneous [41], with a subset of astrocytes appearing hypertrophic, another directing processes towards the lesion and a distinct subset undergoing limited proliferation at juxtavascular sites. Given the intimate association of astrocytes with neurons and other CNS cell types, it is highly likely that any morphological rearrangement which affects this exquisite interaction will impact on CNS function [1]. Perhaps unsurprisingly, cellular hypertrophy in reactive astrocytes also leads to the redistribution of integral membrane proteins. In the APP/PS1 mouse model of Alzheimer’s type amyloidosis there is a redistribution of the bestrophin1 (Best1) channel from astrocyte processes to the cell soma, which is concomitant with increased GABA release from astrocytes and reduced firing probability in granule cells [121]. It is likely that aberrant synaptic function is one of the first events in chronic neurodegenerative diseases, such as Alzheimer’s, which are ultimately characterized by synapse loss and neuronal death. Interestingly, there are indications that in humans astrocyte dysfunction occurs first, with uptake of a PET tracer for monoamine oxidase B (MAO-B), a critical enzyme in the GABA synthesis pathway, positively correlating to uptake of the amyloid marker Pittsburgh Compound B [122]. In this case, however, it needs to be remembered that MAO-B is also expressed in serotonergic neurons, which may influence data interpretation, despite serotonergic neurons comprising a comparatively small population of cells in the adult human brain.

Injury and disease also promote massive changes in gene expression levels, with some genes increasing by over 100-fold [123]. Consistent with the well-established concept of inter-regional astrocyte heterogeneity, TRAP experiments in a common mouse model of multiple sclerosis (experimental autoimmune encephalomyelitis) showed profound differences in gene expression between spinal cord, cerebellum, cerebral cortex and hippocampus in the disease state [124]. Whole transcriptome sequencing of FACS isolated astrocytes from mouse models of bacterial infection (lipopolysaccharide, LPS) [123] and stroke (middle cerebral artery occlusion, MCAO) [123] indicated that astrocytes have profoundly different responses depending on insult. Generally speaking, LPS treatment results in the upregulation of genes typically associated with a pro-inflammatory response, including components of the innate immune system (complement cascade) and pro-inflammatory cytokines. In contrast, MCAO generally results in the enhanced expression of genes associated with an anti-inflammatory response, including neurotrophic factors and anti-inflammatory cytokines. This has led to the introduction of two classes for reactive astrocytes, the so-called pro-inflammatory A1 type and neuroprotective A2 type. This classification is based on the concept of M1/M2 polarization in macrophages [125]. Pro-inflammatory A1 astrocytes are toxic to mouse neurons in co-cultures and have been identified in human samples from patients suffering from AD, ALS, Parkinson’s disease (PD) and multiple sclerosis [126], suggesting that neuronal loss in these diseases results, at least in part, due to reactive astrogliosis. Although the A1/A2 framework is conceptually useful, it should be remembered that the original M1/M2 classification has since proved to be problematic, as it represents the ends of a continuum which rarely, if ever, exists in vivo [125]. Consistent with this, a recent study of transcriptional changes in human caudate samples of Huntington’s disease found little evidence for a clear A1 or A2 phenotype [127], with evidence for both inflammatory states suggesting astrocyte response during human disease is a highly complex process.

The degree to which astrocytes can discriminate between insults, however, remains unclear at this point. For example, whether astrocytes can distinguish between subtle conformational differences in toxic protein species [128], including amyloid beta in AD [129], tau in AD [130] or α-synuclein in PD [131] remains to be determined, as well as whether this initiates a specific cellular response. However, it does appear that response is graded to the insult. Astrocyte responses to both LPS and MCAO appear variegated within tissue, with reactive astrocytes highly expressing *Lcn2* and *Serpina3n* interspersed with quiescent astrocytes [123]. In the APP/PS1 mouse model of Alzheimer’s, astrocytes close to amyloid beta plaques show upregulation of MAO-B. Along with redistribution of the Best1 channel (see earlier), MAO-B upregulation is thought to underlie the abnormal circuit firing in this model observed at early stages of amyloidosis, due to the aberrant release of GABA [121]. Responses may also be affected by the age of an animal, as TRAP experiments in aged mice revealed loss of synaptic support functions (such as cholesterol production) and a switch towards a pro-inflammatory state [50,54]. As these changes appear mirrored in the aging human brain [55], and aging is the single biggest risk factor for neurodegenerative disease, this raises the question of how chronic low-level inflammation contributes to these conditions. Whether differences in astrocyte response also explain regional susceptibility to disease, such as seen with the substantia nigra in PD, also remains unclear [132].

Toxins which build up in the CNS need to be efficiently removed, but the CNS lacks a traditional lymphatic system, which usually serves in such a capacity. Recently, however, Nedergaard and colleagues have advanced the concept of the glia-lymphatic (glymphatic) system, a highly polarized transport system for cerebrospinal fluid (CSF) and interstitial fluid (ISF) in the brain that facilitates extracellular waste removal through a network of astrocyte-supported perivascular or perineural channels that drain into the cervical and basal meningeal lymphatic networks or the major dural sinuses [133]. Performance of the glymphatic system is impaired with aging [134] and may be a contributory factor to the development of chronic neurodegenerative disease. In fact, impairment of glymphatic function adversely affects amyloid beta removal in the APP/PS1 mouse model, suggesting that the system is a critical modulator of AD progression [135]. Crucially, impaired glymphatic function has also been shown in animal models of traumatic brain injury [136] and small ischemic lesions [137], suggesting that toxin build up, due to inefficient removal, plays a significant secondary role in tissue damage, following the initial trauma. The impact of astrocyte heterogeneity on the efficiency of this system, and whether it contributes to regional susceptibility to disease, also needs to be explored. However, two observations point towards the glymphatic system being a general system of waste removal. First, efficient toxin removal is central to CNS function and survival. Second, the astrocyte-specific water channel Aquaporin4 (Aqp4) is central to the system and is one of the few markers expressed across all astrocytes in major single cell sequencing studies performed to date [2,3,63].

Whether astrocytes can eventually revert to their pre-injury state is unclear, but likely depends on insult. For example, although most gene expression changes induced by LPS or MCAO are transient, several genes remain elevated one week after the insult [123] and this may have functional consequences. A better understanding of the changes occurring in reactive astrocytes may lead to interesting avenues for brain repair following injury or disease. It appears a limited number of astrocytes in the mouse striatum and medial cortex can reactivate a Notch-dependent latent neurogenic program following a stroke, resulting in the generation of new neurons [138]. Crucially, however, even though this intrinsic repair seems limited, it does appear that the reactive astrocyte state could be harnessed for brain repair. Reactive cortical astrocytes in the 5xFAD Alzheimer’s model appear to be more easily converted into neurons by retroviral-mediated overexpression of the neurogenic transcription factor NeuroD1 [139], as do reactivate astrocytes generated by a cortical stab wound injury [139]. These induced neurons possess Na^+^ and K^+^ currents, generate action potentials and show spontaneous and evoked responses, arguing for integration into local circuits. Despite the obvious potential of such a system, it remains unclear why the neurons generated by NeuroD1 reprogramming are mainly glutamatergic and express markers typically found in neurons in deep cortical layers [139]. Whether this will impact full functional recovery needs to be systematically assessed across injuries. Understanding the interconversion process, including whether injury causes partial dedifferentiation of astrocytes into a more ‘stem-like’ state [52], should prove informative and identify specific transcriptional pathways that can be targeted for production of specific neuronal types. A further limitation of the system appears to be the resistance of white matter astrocytes to reprogramming [140], which needs to be considered as primates possess more white matter than rodents.

As evidence for discrete subsets of human astrocytes emerges with specific functions, in both healthy [141] and diseased tissues [60], it seems likely that similar concepts of heterogeneous and graded response, age dependency, as well as waste removal and potential for interconversion, will be found.

## 8. Conclusions

Based on a number of criteria, astrocytes have emerged as far more heterogeneous than previously thought and are critically importantly for formation and function of the healthy CNS. Perturbed function of specific astrocyte subsets can lead to disease, while astrocyte response to external injury and/or toxic insults is also heterogeneous. Failure to understand and modify astrocyte response to re-establish CNS homeostasis may explain the continued failure of CNS drugs to reach the clinic, and represents a currently undervalued concept in therapeutics development.

Ultimately, continued improvements in tool development, allowing the selective labeling and manipulation of specific astrocyte subsets, will be key to furthering our understanding on how astrocyte subtypes contribute to CNS development and function, both in the normal and pathological CNS.

## Figures and Tables

**Figure 1 brainsci-10-00168-f001:**
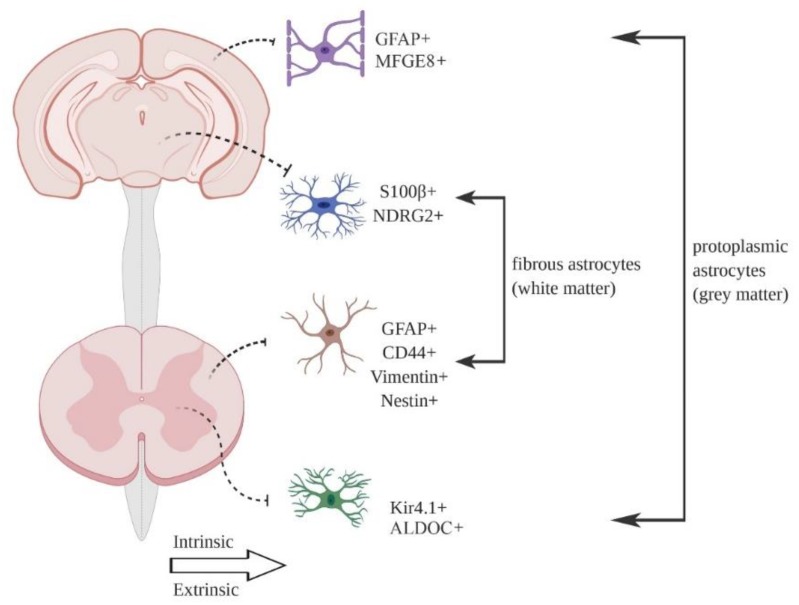
Astrocyte heterogeneity in the central nervous system (CNS). Different populations of astrocytes have been identified in the grey and white matter of the rodent brain and spinal cord, based on differences in morphology and marker protein expression. Astrocyte heterogeneity may arise in two separate ways. During development, astrocytes are generated from distinct pools of progenitor cells, which express unique combinations of transcription factors that drive cell identity (intrinsic). Alternatively, astrocyte diversity may be driven by cues from neighboring cells (such as sonic hedgehog release) (extrinsic) in adult tissue. In reality, it is likely that a combination of intrinsic and extrinsic factors drives astrocyte heterogeneity (see the main text for details).

**Figure 2 brainsci-10-00168-f002:**
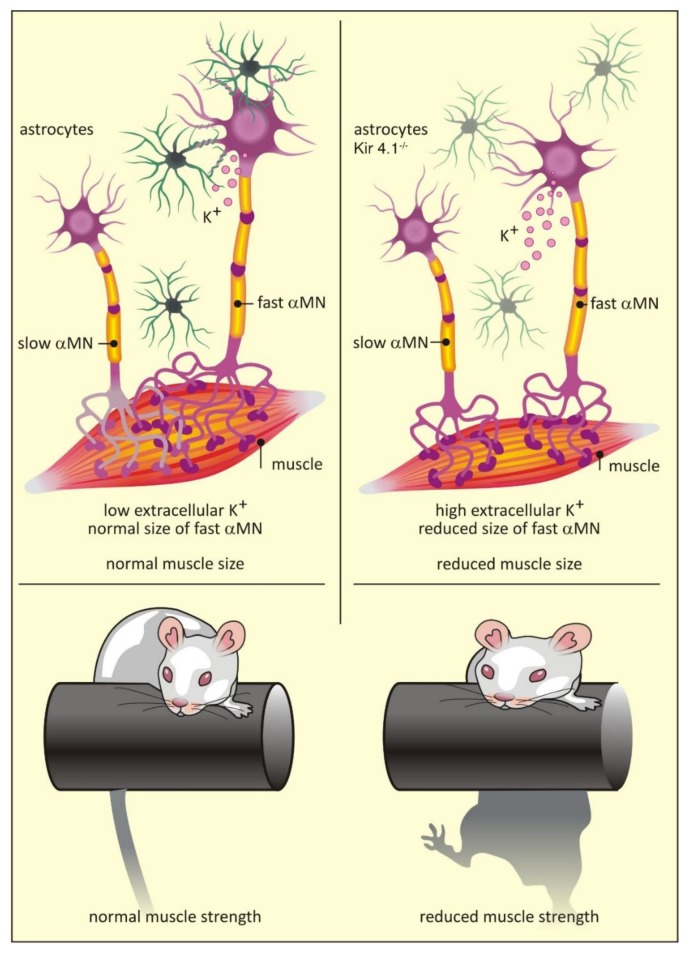
Transcriptome impacts function: the consequences of astrocyte heterogeneity. Differences in transcript expression between astrocytes can affect the function of local neuronal circuits, ultimately leading to functional impairments. For example, astrocytes in the ventral horn of the spinal cord are enriched in expression of *Kcnj10*, which encodes the K^+^ channel Kir4.1. Astrocyte-specific deletion of Kir4.1 leads to impaired local K^+^ homeostasis, a selective decrease in fast α-motor neuron (MN) size and impaired function, a decrease in the size of the innervated muscle and consequent loss of peak strength. Figure is adapted from Kelly et al. [47].

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
