# Peer review of "No Longer Underappreciated: The Emerging Concept of Astrocyte Heterogeneity in Neuroscience"

_brainsci, 2020, doi:10.3390/brainsci10030168_

Round 1

Reviewer 1 Report

The review manuscript submitted by Pestana and collaborators, intends to summarize the current understanding of astrocyte heterogeneity.
The review is generally well written and gives a very good impression. The idea of focusing on astrocyte heterogeneity is interesting, timely and the topic is very interesting indeed. At the same time this review needs more effort to add fresh and original angle to this rapidly developing field.

I have couple of major suggestions/comments:

as it stands, the manuscript is a review of the current literature, and many of the current beliefs, but it lacks any significant interpretation by the authors. Given the standing of the authors in the field, I would have been expecting more of an ‘opinion leader’ piece, which is critical of various experiments/outcomes – and discusses the major challenges that lie ahead for the field The section “Functional Heterogeneity” is lacking some of the new evidence showing that a group of cortical astrocytes can regulate dopamine homeostasis.  The section “Heterogeneity in Injury and Disease”: functional and molecular heterogeneity of astrocytes in normal brain has just been described and whether the disease astrocytes share similar heterogeneity as observed in normal condition is unclear. Indeed, how different reactive astrocytes are involved in different neurologic pathology remains to be largely unknown and need to be further understood. A significant amount of specificity is needed to address this issue.

Author Response

We thank the reviewer for the overall positive assessment of the manuscript. We have taken note of the suggestions, see below, and have attempted to implement all requested changes.

  • As it stands, the manuscript is a review of the current literature, and many of the current beliefs, but it lacks any significant interpretation by the authors. Given the standing of the authors in the field, I would have been expecting more of an ‘opinion leader’ piece, which is critical of various experiments/outcomes – and discusses the major challenges that lie ahead for the field

Obviously, trying to cover a range of aspects in a ‘primer’ type format is difficult. However, we accept the reviewer’s comment and, where relevant, have added text describing experimental limitations and major challenges still to be addressed in the field.

  • The section “Functional Heterogeneity” is lacking some of the new evidence showing that a group of cortical astrocytes can regulate dopamine homeostasis.

We have amended the ‘Functional Heterogeneity’ section, so that it now includes mention of the relevant work.

  • The section “Heterogeneity in Injury and Disease”: functional and molecular heterogeneity of astrocytes in normal brain has just been described and whether the disease astrocytes share similar heterogeneity as observed in normal condition is unclear. Indeed, how different reactive astrocytes are involved in different neurologic pathology remains to be largely unknown and need to be further understood. A significant amount of specificity is needed to address this issue.

We thank the reviewer for this comment, which forced us to think deeply about the structure and overall aim of the manuscript.

We agree that the issue of how different reactive astrocytes are involved in various neuropathologies is unknown and needs to be further understood. However, as it appears likely that all injuries and diseases elicit different responses, we feel it would be impossible to describe in detail all conditions within the structure of the current review. A thorough description of astrocyte responses to each injury/disease would presumably require a separate review, as seen for example with Alexander’s Disease (Messing and Goldman, J Neurosci, 2012), Ischemic Injury (Takano and Nedergaard, Stroke, 2008) and Alzheimer’s Disease (Perez-Nievas and Serrano-Pozo, Front Aging Neurosci, 2018).

Rather, our aim was to illustrate emerging concepts of heterogeneity in response to injury and disease using pertinent examples.

However, we do accept the point of the reviewer and have made significant changes to the section, which we feel maintain the overall goal of the manuscript to act as a ‘primer’ on astrocyte heterogeneity but which now contains specific details of the cited studies.

Finally, we have added additional text at the beginning of the section clarifying our aim, whilst making clear that more comprehensive discussion of all aspects of astrocyte response to a specific injury or disease can be found in dedicated review articles.

We hope that by explaining our approach, and making appropriate changes to the text, that we have succeeded in satisfying the reviewer.

Reviewer 2 Report

In this review article, Pestana et al., summarized the recent findings about astrocyte heterogeneity based on several characteristics of astrocytes. Overall, the content and the flow of the manuscript are acceptable.  

However, references are not placed in the right places. It is hard to find which reference is cited in particular sentences. Please improve it throughout the manuscript. Especially, since authors cited their two papers that are recently accepted, and they are not available. I cannot evaluate properly whether their conclusions are accurate or correct.

There are several odd expressions or sentences throughout the manuscript. Please reconsider them for modification.

Line 19: traditionally studied as a homogeneous group of cells,

Line 53: accumulating supporting?

Line 81: Astrocyte heterogeneity is driven by region restricted progenitors utilizing distinct transcription factor codes

Line 122: proof? --> /the proof?

Line 143: Wnts à What is Wnts? Wnt signaling or Wnt ligands?

Line 146: Emx1 à What is Emx1?

Line 222: concern moving forward?

Line 239: shotgun sample all cells?

Line 263: can seem impenetrable?

Line 383: affect the functional? Of local neuronal circuits. Leading to disease? (diseases?)

Line 416: long-term it has?
